# A cortical cell ensemble in the posterior parietal cortex controls past experience-dependent memory updating

Akinobu Suzuki [1,2,3], Sakurako Kosugi[1,2], Emi Murayama[1,2,3], Eri Sasakawa[1], Noriaki Ohkawa [1,2,4,6], Ayumu Konno [5], Hirokazu Hirai [5] & Kaoru Inokuchi [1,2,3 ✉]

When processing current sensory inputs, animals refer to related past experiences. Current information is then incorporated into the related neural network to update previously stored memories. However, the neuronal mechanism underlying the impact of memories of prior experiences on current learning is not well understood. Here, we found that a cellular ensemble in the posterior parietal cortex (PPC) that is activated during past experience mediates an interaction between past and current information to update memory through a PPC-anterior cingulate cortex circuit in mice. Moreover, optogenetic silencing of the PPC ensemble immediately after retrieval dissociated the interaction without affecting individual memories stored in the hippocampus and amygdala. Thus, a specific subpopulation of PPC cells represents past information and instructs downstream brain regions to update previous memories.

[1] Department of Biochemistry, Graduate School of Medicine and Pharmaceutical Sciences, University of Toyama, 2630 Sugitani, Toyama 930-0194, Japan. [2] CREST, JST, University of Toyama, Toyama 930-0194, Japan. [3] Research Center for Idling Brain Science, University of Toyama, Toyama 930-0194, Japan. [4] PRESTO, JST, 4-1-8 Honcho, Kawaguchi, Saitama 332-0012, Japan. [5] Department of Neurophysiology and Neural Repair, Gunma University Graduate School of Medicine, Maebashi, Gunma 371-8511, Japan. [6]Present address: Division for Memory and Cognitive Function, Research Center for Advanced Medical Science, Comprehensive Research Facilities for Advanced Medical Science, Dokkyo Medical University, Tochigi 321-0293, Japan. ✉email: inokuchi@med.u-toyama.ac.jp

Animals compare current experiences, either implicitly or explicitly, with prior experiences[1]. New information is then incorporated into the related neural network to update previously stored memories. Memories are stored in neuronal subpopulations called engrams or memory traces that are activated during experience[2]. Memory is then updated through a dynamic process depending on the new situation. For example, original memories can be updated with new information (integration and association)[3] or altered by opposite information (extinction and counterconditioning)[4] for survival and higher-order function. Recent studies have begun to clarify the mechanisms underlying memory engrams and how prior experiences affect later learning. Memories encoded close in time interact to generate an associative memory or facilitate the encoding of the later event[2,5]. Sharing of engram cell populations mediates the interaction between the prior and the later memories[6–10]. These studies have focused on engrams in limbic system structures such as the hippocampus and amygdala, which are primary storage sites of engrams. Human studies have indicated that a cortical modulation system controls the active integration of new information into the neural network that represents past information[3,11,12]. Thus, cortical top-down modulation may instruct engrams stored in the downstream regions to interact to update memory.

The posterior parietal cortex (PPC) is a major associative region in the mammalian cortex. The PPC is classically considered to be associated with cognitive functions, such as visual perception[13], spatial attention and visual guidance[14,15], and episodic memory in humans and rodent[16–18]. Recent findings indicate that the PPC plays important role in memory updating. The PPC modulates fear memory renewal after the extinction in fear conditioning[19]. The PPC represents past sensory history and influences behavioral outcomes upon current experience[20]. Furthermore, the PPC implements updating of the prediction when the uncertainty of the prediction decreases with new sensory inputs[21]. However, an important and outstanding question is how representations of past experiences in the PPC interact with current sensory inputs. The complexity of the experimental design used in the previous studies makes it technically difficult to identify the specific cellular populations and assign them to past or current experience. In this study, by taking advantage of a simple behavioral task in which mice associate a past experience (context) with a current sensory input (footshock) during one learning session, we show that a specific subpopulation of cells in PPC governs past event-dependent memory update. The PPC cell ensemble has characteristics distinct from the engram cell ensemble as it does not encode information of individual events, rather it controls the interaction between past and current memory events.

## Results

**Neuronal ensemble in PPC responds to past and current experiences.** In a context pre-exposure and immediate shock (IS) task, animals retrieve a previously encoded memory of the pre-exposed context, which is then associated with the current experience, an IS in the same context, to update memory, such that animals exhibit a fear response in this context[22,23]. This is followed by an increase in the degree of overlapping in engram cell populations both in the hippocampus and amygdala[23]. This is a simple learning task, and so is suitable for detecting the corresponding neuronal ensembles responsible for memory updating.

We found that an IS without pre-exposure failed to associate the context and shock (Fig. 1a, b). Conversely, mice showed higher freezing in test sessions when pre-exposed and IS contexts

were the same (paired) than distinct (unpaired) contexts, irrespective of the interval between context pre-exposure and the IS (Fig. 1b, c). Cellular compartment analysis of temporal activity by fluorescence in situ hybridization (CatFISH) utilizing an immediate early gene Arc permits the detection of activity-dependent activation of large neuronal ensembles during two distinct events[24]. After neuronal activation, Arc RNA is initially localized to the nucleus but then translocates to the cytoplasm within 30 min. Therefore, the expression of cytoplasmic and nuclear Arc RNA can be used to distinguish neuronal populations engaged by the behavioral experience of the first event (pre-exposed context) from those engaged by the behavioral experience of the second event (IS). CatFISH revealed that, in addition to the hippocampus and amygdala, several cortical regions responded to both events (Supplementary Fig. 1). The neuronal population of cytoplasmic and nuclear Arc double-positive cells in the PPC of the paired group was significantly larger than that in the unpaired and No IS groups (Fig. 1d–f). This suggests that the neuronal population of double-positive cells in the PPC responds to past and current experiences. The degree of overlap in each group was significantly higher than the corresponding chance level. Moreover, the degree of Arc double-positive cells of the No IS group was significantly larger than that in the unpaired group. This suggests that memory updating might take place even when the behavioral phenotype does not change significantly.

**Optical silencing of the neuronal ensemble in PPC blocks memory association.** We used c-fos-tTA mice that express the tetracycline-transactivator (tTA), which is under the control of c-fos promoter, an immediate early gene used as a marker of neuronal activity (Fig. 2a). In the absence of doxycycline (Dox), activation of the c-fos promoter leads to tTA expression, and tTA binding to the tetracycline-responsive element (TRE) induces target gene expression. In the presence of Dox, tTA is prevented from binding to the TRE. We found that context pre-exposure induced tTA mRNA expression in the PPC (Supplementary Fig. 2). A recombinant lentivirus (LV) encoding TRE-ArchT-EYFP or TRE-EYFP was injected bilaterally into the PPC of c-fos-tTA mice to label cells that were activated during context pre-exposure (Fig. 2b and Supplementary Fig. 3). LV-injected mice were taken off Dox (OFF Dox group) for 2 days and then exposed to context B to label ArchT-EYFP or EYFP proteins in neurons that were activated during pre-exposure. The next day, mice received an IS in the same context while optical illumination (continuous 589 nm light) was delivered to the PPC during the IS session (Light ON). The function of ArchT-EYFP was restricted to the Light ON/OFF Dox group; the IS session-induced expression of Zif268 (also called Egr1), an immediate early gene, in the PPC was specifically suppressed in this group (Supplementary Fig. 4). The other groups were treated in the same way. The ArchT-Light ON group showed significantly lower freezing in the test session than the ArchT-Light OFF and EYFP-Light ON groups (Fig. 2b). These results indicate that a PPC neuronal population is required for the association between the pre-exposed context and IS. A previous study indicates that the PPC lesion does not affect standard contextual fear conditioning, in which shock was delivered at the end of the context exposure[25], suggesting that the PPC is not necessary for CFC when there is no updating. This result supports our finding that the PPC specifically regulates past event-dependent memory update.

Furthermore, the silencing of PPC cells that were activated in a different context had no effect on fear expression in the pre-exposed context. PPC cells, which were activated during pre-exposure to the circle context, were labeled with ArchT (circle-labeled group). Mice

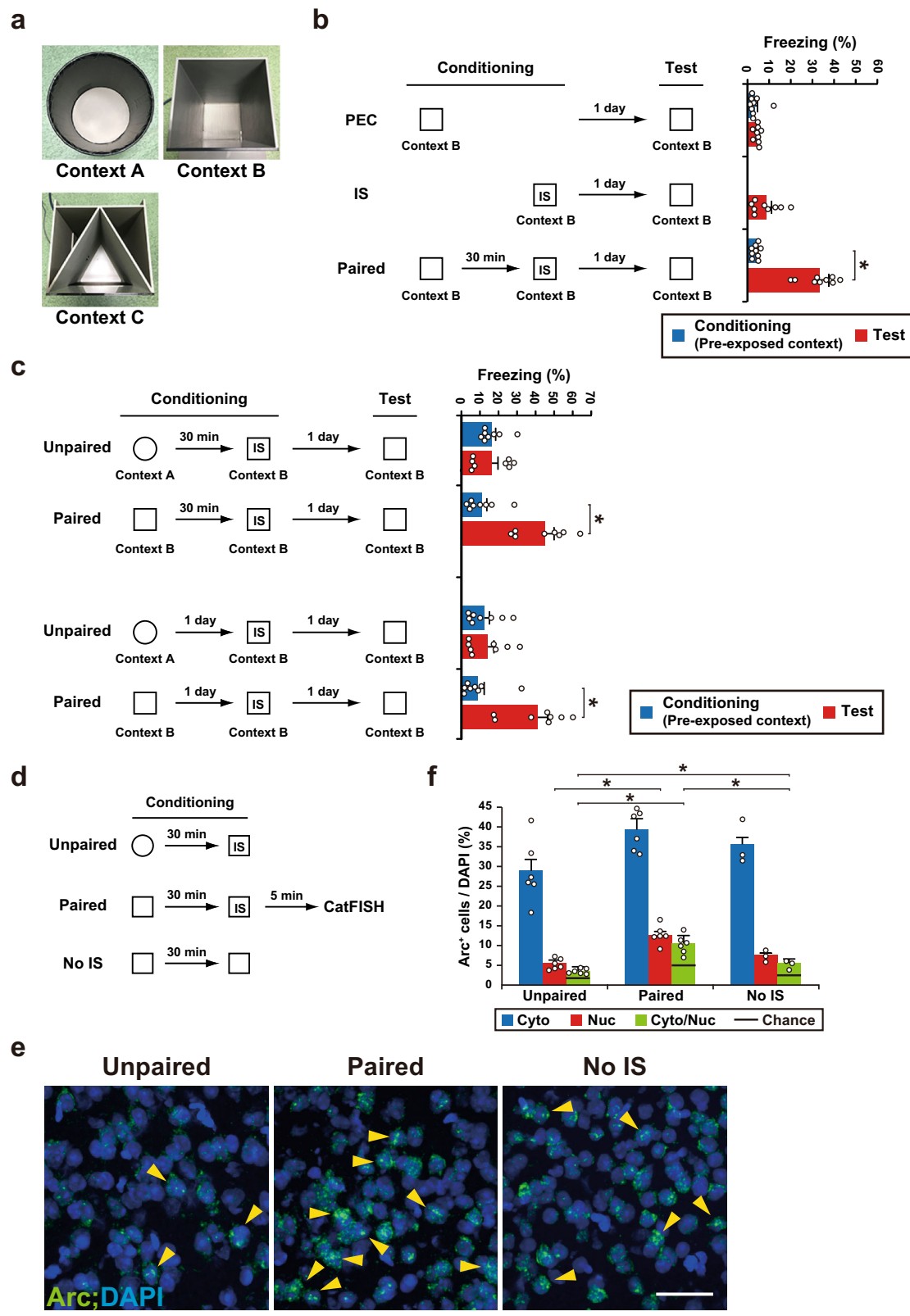

received the paired condition of the CPFE paradigm with optical silencing to the PPC during an IS session in the square context. The circle-labeled group showed significantly higher freezing in the test session even when optical silencing was delivered to the PPC cell population that responded to the circle context (Fig. 2c). This indicates that cell populations labeled in the PPC are specific to each context.

Optical silencing of the PPC cells activated during the context pre-exposure had no effect on the retrieval of pre-exposed context memory (the ArchT-Light ON group), which was evident from the finding that motility, an indicator of contextual memory[26], was comparable to other groups (Fig. 2d). When the same strategy was applied to the hippocampal CA1 region, the ArchT-Light ON group showed significantly higher motility than the

**Fig. 1 PPC cell ensembles corresponding to the pre-exposed context and IS. a** Photo showing context A–C used in this study. **b** Schema of pre-exposed context (PEC), immediate shock (IS), and PEC-IS paired paradigm. The graph shows the freezing levels during the conditioning and test session ($n = 9$ mice/group) (conditioning; two-tailed unpaired $t$-test, $P = 0.8047$, test; one-way ANOVA followed by Tukey's post hoc tests, session: $P < 0.0001$). **c** Schema of the unpaired and paired paradigm. Mice were pre-exposed to context A (unpaired group) or context B (paired group), and then received an IS in context B 30 min or 1 day later. Mice were then tested 1 day after the conditioning. The graph shows the freezing level observed during the conditioning and test session (interval between Ctx and IS: 30 min: $n = 8$ mice/group; 1 day: $n = 8$ mice/group) (two-way ANOVA with repeated measures (RM) followed by Sidak's post hoc test, interval between Ctx and IS; 30 min: session: $P < 0.0001$, group: $P = 0.0223$, session × group: $P < 0.0001$, 1 day: session: $P = 0.0003$, group: $P = 0.0211$, session × group: $P = 0.0008$). **d** Schema of Arc CatFISH analysis. Three groups (unpaired, paired, and the no IS group, which did not receive a footshock) were used in this study. Mice were sacrificed for Arc CatFISH analysis 5 min after the conditioning. **e** Representative images of the PPC CatFISH analysis. The Arc RNA signal and DAPI nuclear staining are shown in green and blue, respectively. The yellow arrowheads indicate Nuclear+, Cytoplasmic+ (double-positive) cells. Scale bar, 50 μm. **f** Percentage of Arc+ cells per DAPI+ cells (unpaired and paired group, $n = 6$ mice, no IS group, $n = 3$ mice). Error bars indicate the mean ± s.e.m. (One-way ANOVA followed by Tukey's post hoc tests, Cyto: $P = 0.0515$, Nuc: $P = 0.0001$, Cyto/Nuc: $P < 0.0001$). Error bars indicate the mean ± s.e.m. *$P < 0.05$. Cyto cytoplasmic, Nuc nuclear. Black line indicates chance. For details of statistical data, see Supplementary Table 1. Source data are provided as a Source Data file.

other groups, which indicates that CA1 acts as a memory storage site of the pre-exposed context (Fig. 2d).

We applied the social interaction test to measure the IS memory recall under the assumption that IS-induced fear would suppress social interaction behavior. Indeed, mice that received IS showed a decrease in interaction time and an increase in latency to first interaction compared with No IS mice (Supplementary Fig. 5). Thus, the social interaction test is appropriate for the assessment of whether or not IS memory was intact. Optical silencing of a specific small subpopulation of basolateral amygdala (BLA) neurons that were activated during the IS and subsequently act as a memory trace for the IS experience (i.e., IS memory cell ensemble)[23] enhanced social interaction behavior, with a longer interaction time and shorter latency compared with the control groups (Fig. 2e). This result strongly suggests that the inhibition of IS memory recall caused the suppression of anxiety, although it is unclear whether random inhibition of BLA neurons would rescue IS-induced anxiety. By contrast, optical silencing of the PPC cells activated by the IS did not affect social interaction, which was not significantly different from the other groups (Fig. 2e). This finding indicates that IS memory was stored in the BLA and IS memory recall was independent of the activity of the PPC cell ensemble. Similar results were obtained when the sodium channel blocker lidocaine was injected into the PPC (Supplementary Fig. 6). This treatment inhibited the pre-exposed context-IS association (Supplementary Fig. 6a) without affecting the proper recall of the pre-exposed context and IS memories (Supplementary Fig. 6b, c; compare to vehicle control).

**Optical activation of the neuronal ensemble in PPC generates an artificial associative memory.** We next examined whether stimulation of the PPC cell ensemble can generate an artificial pre-exposed context-IS associative memory (Fig. 3). We applied an unpaired paradigm in which mice did not associate the pre-exposed context with the IS. ChR2-EYFP functioned in the PPC cell ensemble specifically in the Light ON and OFF Dox condition, which was judged by endogenous c-Fos induction (Fig. 3a and Supplementary Fig. 7). The c-fos-tTA mice who received LV vector injections into PPC were in the OFF Dox condition for 2 days, and then pre-exposed to context A to label the activated PPC cells. One day later, mice received an IS in the different context B while 20 Hz light pulses were delivered to the PPC. The ChR2-Light ON group exhibited a higher freezing response in the pre-exposed context (context A), in which mice did not receive a footshock than the other groups (Fig. 3b). The freezing response of the ChR2-Light ON group in the neutral context (context C) was comparable to that of the control group, which indicates that optically induced artificial memory is context-specific.

**The PPC regulates associative memory retrieval.** Suppression of the PPC ensemble activity transiently dissociated the pre-exposed context-IS association that had already been stored in the brain (Fig. 4a, b). A subset of the PPC cell ensemble that responded to memory reactivation was labeled with ArchT-EYFP. Optical silencing of the PPC ensemble during test 1 suppressed memory recall. However, the suppressed memory returned to the control level the next day (test 2).

**Optical silencing of PPC ensemble activity dissociates the memory association.** A previous study indicated that optical inhibition of CA1 neurons for 15 min immediately after memory reactivation leads to disruption of fear memory[27]. Consistent with previous evidence, optical silencing of PPC neurons immediately after test 1 suppressed the fear memory recall 1 day later in test 2 (Fig. 5a, b), presumably via a reconsolidation inhibition-like mechanism. In the other groups, optical silencing had no effect on recall of pre-exposed context (Fig. 5c, motility, test 2) or IS (Fig. 5d, vehicle, test 3) memories. When CA1 activity was suppressed by lidocaine injection, mice showed higher motility in test 3 than vehicle-injected mice (Fig. 5c). Similarly, BLA suppression resulted in a longer interaction time and shorter latency of the first interaction (Fig. 5d, test 3, lidocaine). Thus, suppression of pre-exposed context-IS associative memory (Fig. 5b) was not due to an impairment in pre-exposed context or IS memory.

**Optical silencing or activation affects c-Fos expression in the BLA and ACC after a test session.** What neural circuit downstream of the PPC regulates pre-exposed context-IS memory association? We found a decrease or increase, respectively, in the number of c-Fos-positive cells when the PPC cells of interest were optically silenced (Fig. 6a, b, e, f; paired condition) or stimulated (Fig. 6c, d; unpaired condition), not only in the BLA but also in the anterior cingulate cortex (ACC), which also plays a critical role in regulating fear responses[28–30].

**PPC–ACC circuit regulates memory association.** The PPC was found to be bidirectionally connected to the ACC by injecting an anterograde tracer, Biotinylated Dextran Amines (BDA), into the PPC and a retrograde tracer, cholera toxin subunit B-Alexa Fluor 488 conjugate (CTB488), into the ACC (Fig. 7a and Supplementary Fig. 8). To examine the involvement of this PPC-ACC circuit in-memory association, we performed a similar experiment to that shown in Fig. 2b, except that AAV9 TRE-ArchT-EYFP was infused directly into the PPC and optical silencing was delivered to the ACC (Fig. 7b and Supplementary Fig. 9). The expression and function of ArchT-EYFP in the ACC were

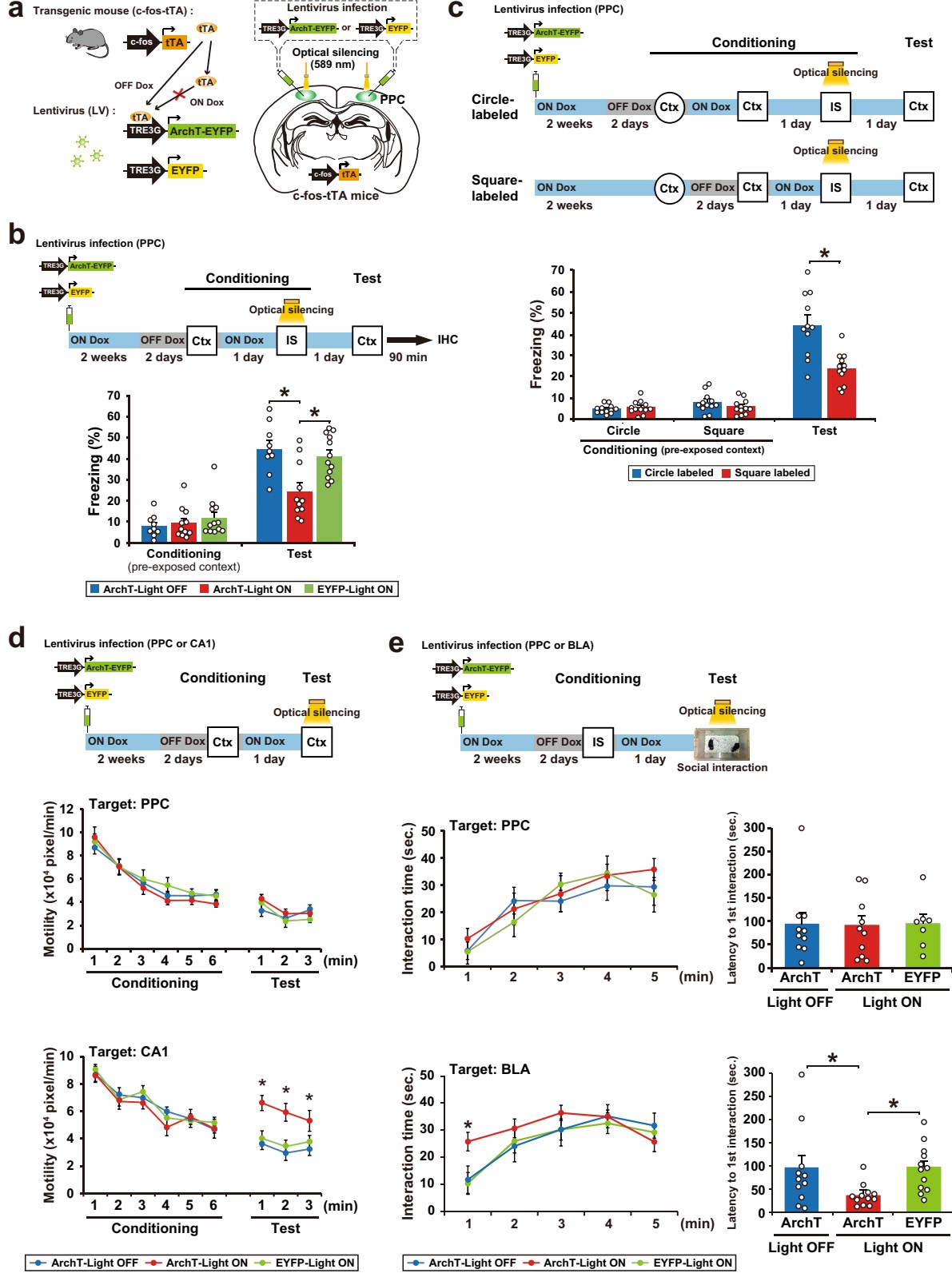

restricted to the Light ON and OFF Dox condition (Supplementary Fig. 10). Optical silencing of the PPC neuronal terminals at the ACC that was delivered during the IS session blocked the formation of memory association (Fig. 7c). This was accompanied by a decrease in the number of c-Fos-positive cells in the BLA and ACC (Fig. 7d), indicating that the PPC–ACC circuit regulates memory updating.

## Discussion

Our study reveals that a PPC cell ensemble recruited during a past experience acts as a top-down modulator for the interaction between recalled memory and ongoing experience. This is consistent with previous findings that the PPC represents past experiences[20] and is involved in episodic memory retrieval[31]. The PPC has been proposed to act as a sensory history buffer, i.e., past

**Fig. 2 Optical silencing of cell assemblies in the PPC blocks memory association without affecting the original memories. a** Labeling of PPC neurons in c-fos-tTA transgenic mice with the LVs TRE3G-ArchT-EYFP or TRE3G-EYFP. **b** Silencing of the cell ensembles during an IS session (top). The blue and gray bars indicate the presence or absence of Dox, respectively. The graph shows the freezing level of each group during the conditioning and test session (bottom) (ArchT-Light OFF, $n = 9$ mice, ArchT-Light ON, $n = 11$ mice, EYFP-Light ON, $n = 12$ mice) (two-way ANOVA with RM followed by Sidak's post hoc test, session: $P < 0.0001$, group: $P = 0.0089$, session × group, $P = 0.0018$). Mice were sacrificed 90 min after the test for immunohistochemistry (Fig. 6a, b). **c** Silencing of the cell ensembles corresponding to circle or square context during an IS session (top). The graph shows the freezing level of each group during the conditioning and test session (bottom) (circle-labeled, $n = 12$ mice, square-labeled, $n = 12$ mice) (two-way ANOVA with RM followed by Sidak's post hoc test, session: $P < 0.0001$, group: $P = 0.0009$, session × group: $P < 0.0001$). **d** Optical silencing of context exposure-activated cell ensembles in PPC or CA1 during the test session (top). The graph shows motility during the conditioning and test sessions in the groups with silencing of cell ensemble in PPC (middle) and CA1 (bottom) (PPC; ArchT-Light OFF, $n = 10$ mice, ArchT-Light ON, $n = 10$ mice, EYFP-Light ON, $n = 7$ mice, CA1; ArchT-Light OFF, $n = 12$ mice, ArchT-Light ON, $n = 12$ mice, EYFP-Light ON, $n = 13$ mice) (two-way ANOVA with RM followed by Sidak's post hoc test, PPC; conditioning; time: $P < 0.0001$, group: $P = 0.3140$, time × group: $P = 0.9077$, test; time: $P = 0.0092$, group: $P = 0.4356$, time × group: $P = 0.5851$, CA1; conditioning; time: $P < 0.0001$, group: $P = 0.3797$, time × group: $P = 0.9397$, test; time: $P = 0.2378$, group: $P < 0.0001$, time × group: $P = 0.8334$). **e** Silencing the IS session-activated cell ensembles in PPC or BLA during the social interaction test (top). The graph shows the interaction time (left) and latency to the first interaction (right) during the test session in the groups with silencing cell ensemble in PPC (middle) and BLA (bottom) (PPC; ArchT-Light OFF, $n = 10$ mice, ArchT-Light ON, $n = 10$ mice, EYFP-Light ON, $n = 7$ mice, BLA; ArchT-Light OFF, $n = 11$ mice, ArchT-Light ON, $n = 12$ mice, EYFP-Light ON, $n = 12$ mice) (interaction time; two-way ANOVA with RM followed by Sidak's post hoc test, PPC; time: $P < 0.0001$, group: $P = 0.5733$, time × group: $P = 0.9098$, BLA; time: $P = 0.1873$, group: $P < 0.0001$, time × group: $P = 0.3515$, latency to interaction; one-way ANOVA followed by Tukey's post hoc tests, PPC; $P = 0.9920$, BLA; $P = 0.00207$). Error bars indicate the mean ± s.e.m. *$P < 0.05$. Ctx context, IS immediate shock, IHC immunohistochemistry. For details of statistical data, see Supplementary Table 1. Source data are provided as a Source Data file.

experience is held in memory for later comparison with present experience[32]. Our findings suggest that the PPC cell ensemble that is activated during past experience (context pre-exposure) acts as a sensory buffer for use in a future relevant experience.

The PPC ensemble identified shares characteristics with the memory engram[2], in that activity in both is generated as a cellular ensemble during the experience. Furthermore, memories can be modified by manipulating the activity of the PPC ensemble, which is similar to previous studies where manipulation of engram cells in the hippocampal and amygdala generates false memories[23,33]. However, our results indicate that the PPC ensemble function is distinct from that of the memory engram because silencing PPC ensemble activity had no effect on the recall of individual memories of context and fear. Thus, the cell ensemble in the PPC does not store pre-exposed context and IS memories but rather regulates the interaction between these memories. Our previous study showed that the hippocampus and BLA ensembles serve as an engram for pre-exposed context and IS, respectively[23]. Moreover, neocortical memory engram is thought to form slowly in systems memory consolidation[34]. Indeed, it has been reported that memory engram cells in the neocortex are generated during learning in a silent form, and then gradually mature to form a functional engram[35,36]. By contrast, we found that PPC ensemble cells quickly become functional after learning, which is consistent with recent findings of a rapid involvement of neocortical areas in memory processing[37,38]. Our findings suggest that these PPC cells may recruit engram cells in the downstream regions, i.e., the hippocampus and amygdala, upon presentation of the context to express fear memory. The immunohistochemical results (Fig. 6) support this notion by showing that manipulation of the PPC ensemble regulates the number of c-Fos-positive cells in the BLA.

We found that the PPC ensemble is required not only for encoding but also for the recall of associated memory. Post-retrieval suppression of the corresponding cell ensemble dissociated the linkage between pre-exposed context and IS without affecting individual memories. Taken together, our findings indicate that the PPC ensemble can flexibly bridge and disconnect different kinds of information, namely the pre-exposed context and IS, that are stored in downstream regions in a top-down manner.

Another important issue is the identity of the PPC-connected brain region that mediates the role of the PPC in memory updating. We found that the PPC has reciprocal connections with the ACC and that optical silencing of the axon terminal of the PPC cell ensemble in ACC during the IS session blocked the association between pre-exposed context and IS memories. Moreover, the PPC ensemble recruited ACC and BLA neurons (Fig. 6). ACC neurons project to the BLA[30], and the PPC has reciprocal connections with the hippocampus via entorhinal and retrosplenial cortices[39]. The medial prefrontal cortex, including the ACC, is involved in encoding novel but related information into existing knowledge in humans and existing schema in rodent[40–43]. Although further study is required, the PPC ensemble may, via the ACC and entorhinal and retrosplenial cortices, instruct downstream regions (hippocampus and amygdala) to integrate newly acquired information into a related neural network that represents past experiences to create semantic memories and schema.

Re-experiencing unwanted traumatic events, i.e., flashbacks is a major symptom of post-traumatic stress disorder that is triggered by stimuli that were present around the traumatic experience, most of which relate in some way to daily events[44,45]. Thus, our findings may inform research that aims to prevent intrusion-based flashbacks in patients with post-traumatic stress disorder.

## Methods

**Animals**. All animal procedures were performed in accordance with the guidelines of the National Institutes of Health (NIH) and were approved by the Animal Care and Use Committee of the University of Toyama. Naïve male C57BL/6J (Japan SLC, Inc., Shizuoka, Japan) and c-fos-tTA transgenic mice (Mutant Mouse Regional Resource Centre, stock number: 031756-MU) aged 10–18 weeks were used for experiments. c-fos-tTA transgenic mice were raised from the time they were fetuses on food containing 40 mg/kg Dox and maintained on Dox pellets, except for the labeling day. All mice were maintained on a 12 h light/dark cycle (lights on 8:00 am–8:00 pm) at 24 ± 3 °C and 55 ± 5% humidity, had ad libitum access to food and water, and were housed in a cage with littermates until surgery.

**Viral constructs**. The pLenti-TRE3G-eArchT3.0-EYFP and pLenti-TRE3G-EYFP were used[8,10]. The pLenti-TRE3G-ChR2(T159C)-EYFP plasmid was constructed using a two-step process. The ChR2(T159C)-EYFP fragment of pLenti-CaMKII-ChR2(T159C)-EYFP was prepared with the BamHI-EcoRI restriction site. The EcoRI sites of the ChR2(T159C)-EYFP fragment were blunt-ended with the Klenow fragment of *Escherichia coli* DNA polymerase I (Takara Bio Inc., Shiga, Japan, 2140A). The resulting ChR2(T159C)-EYFP fragment was subcloned into pLenti-TRE3G-IRES at the BglII-EcoRV restriction sites, generating a pLenti-TRE3G-ChR2(T159C)-EYFP plasmid. Then, the TRE3G-ChR2(T159C)-EYFP fragment was prepared with EcoRI-BamHI restriction sites. The BamHI sites of the TRE3G-ChR2(T159C)-EYFP fragment were blunt-ended. The resulting TRE3G-ChR2(T159C)-EYFP fragment was subcloned into the EcoRI-EcoRV restriction site

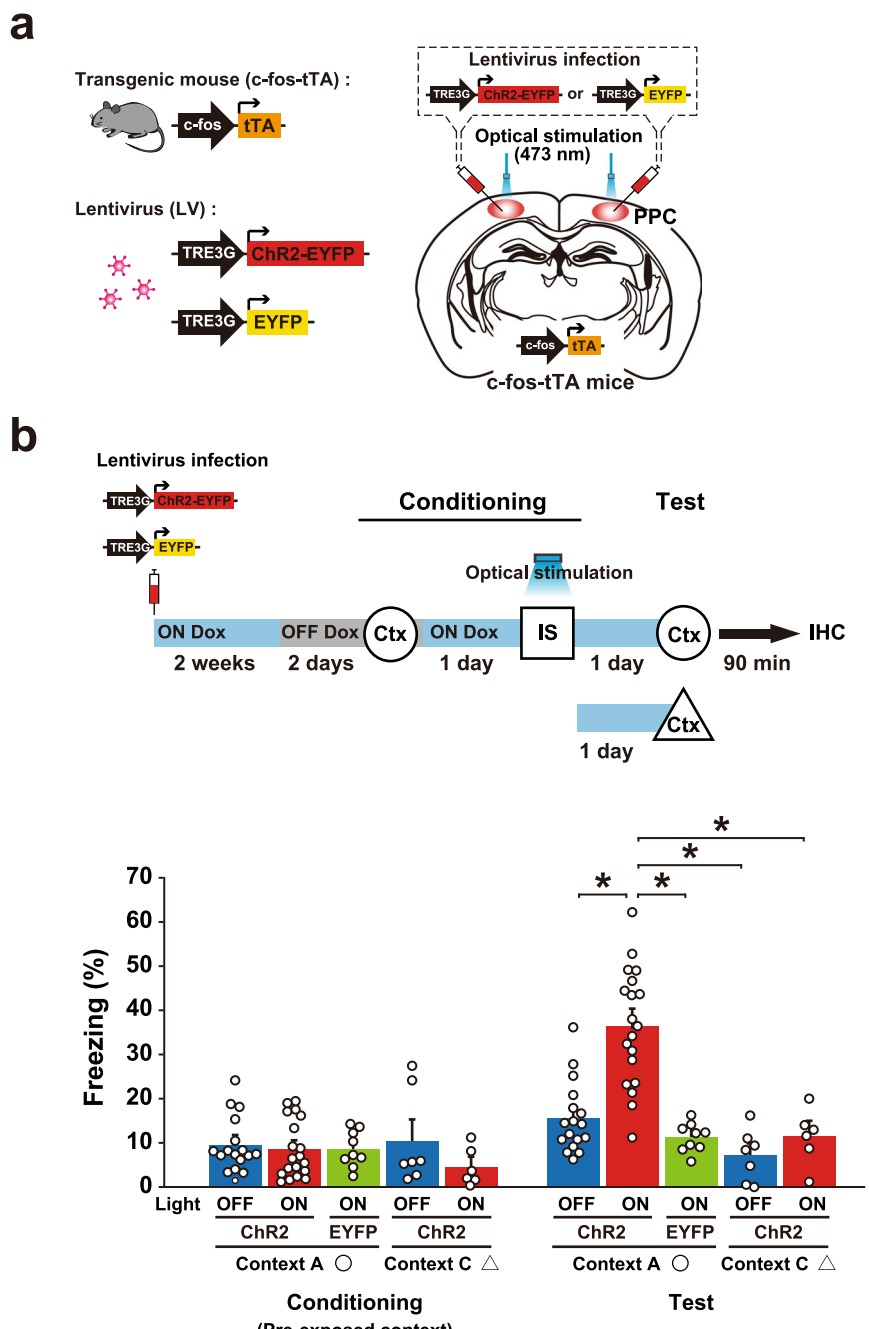

**Fig. 3 Optical activation of the PPC cell ensemble generates an artificial pre-exposed context-IS associative memory. a** Labeling of the PPC neurons in c-fos-tTA transgenic mice with the LVs TRE3G-ChR2-EYFP or TRE3G-EYFP. **b** Activating cell ensembles during an IS session of conditioning (top). The blue and gray bars indicate the presence or absence of Dox, respectively. The graph shows freezing level in context A or C in the conditioning and test session (bottom) (ChR2-Light OFF in circle, $n = 17$ mice, ChR2-Light ON in circle, $n = 19$ mice, EYFP-Light ON in circle, $n = 9$ mice, ChR2-Light OFF in triangle, $n = 7$ mice, ChR2-Light ON in triangle, $n = 6$ mice) (two-way ANOVA with RM followed by Sidak's post hoc test, session: $P < 0.0001$, group: $P < 0.0001$, session × group: $P < 0.0001$). Mice were sacrificed 90 min after the test for immunohistochemistry (Fig. 6c, d). Error bars represent the mean ± s.e.m. *$P < 0.05$. Ctx context, IS immediate shock, IHC immunohistochemistry. For details of statistical data, see Supplementary Table 1. Source data are provided as a Source Data file.

of the TGB plasmid[46], generating a pLenti-TRE3G-ChR2-EYFP plasmid. The above lentiviral plasmids were prepared using an EndoFree Plasmid Maxi kit (Qiagen, Hilden, Germany) and were used for the LV preparation as described previously[23]. The 293FT cells (Invitrogen) were maintained in maintenance medium (Dulbecco's modified Eagle's medium; Gibco, 11995) containing 10% heat-inactivated fetal bovine serum, 2 mM ʟ-glutamine (Gibco, 25030-149), 0.1 mM MEM non-essential amino acids (Gibco, 11140-050), 500 µg/ml geneticin (Gibco, 10131-035), and 1% 100× penicillin/streptomycin (Gibco, 15140-148). Twenty micrograms of the pLenti plasmid and 48 µg of ViraPower Packaging Mix

(Invitrogen, K4975-00) were combined in 6 ml of Opti-MEM (Invitrogen, 31985). The DNA mixture was mixed gently with 6 ml of Opti-MEM containing 144 µl of Lipofectamine 2000 (Invitrogen, 11668). After 20 min, the transfection mix was added to a 225 cm2 flask (BD Falcon, 353138) containing 95% confluent 293FT cells cultured in 28 ml of maintenance medium without geneticin. After 16 h, spent medium was replaced with 30 ml of the virus production medium UltraCULTURE (Lonza, 12–725F) supplemented with 4 mM L-glutamine, 2 mM GlutaMAX-I (Gibco, 35050-061), 0.1 mM minimum essential medium non-essential amino acids, 1 mM sodium pyruvate (Gibco, 11360-070), and penicillin/

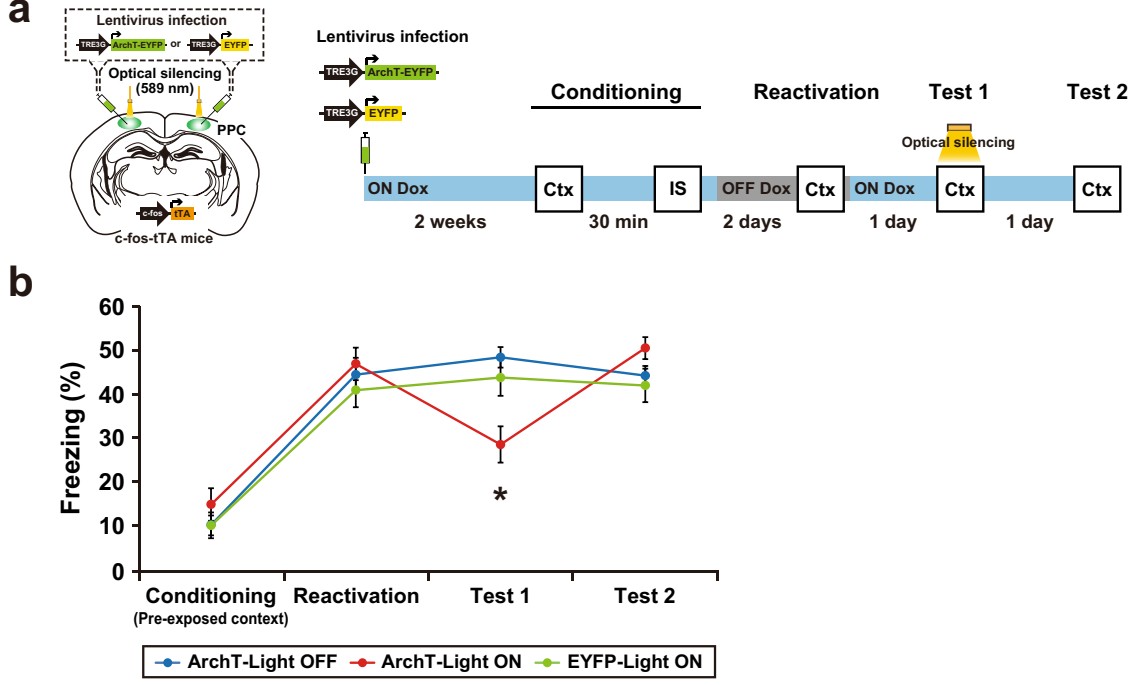

**Fig. 4 The PPC regulates associative memory retrieval. a** Schematics showing labeling of the PPC neurons in c-fos-tTA transgenic mice with the LVs TRE3G-ArchT-EYFP or TRE3G-EYFP (left). Schematic of the behavioral experiment with optical silencing (right). Blue and gray bars indicate the presence or absence of Dox, respectively. Optical silencing to the PPC was delivered during test 1. **b** The graph shows the freezing level during the conditioning, reactivation, test 1, and test 2 (ArchT-Light OFF, $n = 14$ mice, ArchT-Light ON, $n = 13$ mice, EYFP-Light ON, $n = 10$ mice) (two-way ANOVA with RM followed by Sidak's post hoc test, session: $P < 0.0001$, group: $P = 0.6419$, session × group: $P = 0.0002$). Error bars represent the mean ± s.e.m. *$P < 0.05$. Ctx context, IS an immediate shock. For details of statistical data, see Supplementary Table 1. Source data are provided as a Source Data file.

streptomycin. Seventy-two hours after transfection, the culture medium was collected, filtered with a Millex-HV (0.45 µm) (Millipore), and transferred to a centrifuge tube (Hitachi, 40PA tube). Three milliliters of 20% sucrose in phosphate-buffered saline (PBS) was gently added to the bottom of each centrifuge tube; the tubes were then centrifuged in a Hitachi P28S swing rotor for 2 h at 50,000 × g. The pellets of TRE-EYFP-LV, TRE-ChR2-EYFP-LV, and TRE-EYFP-LV were resuspended in 25 and 12.5 µl of cold PBS (Gibco, 14190) per tube, respectively, and the virus solution was transferred to an Eppendorf tube and centrifuged at 5000 × g for 5 min. The supernatant was collected. The viral titers were approximately $5 \times 10^9$–$1 \times 10^{10}$ IU/ml.

The pAAV-TRE3G-ArchT-EYFP plasmid was constructed using the following steps. The TRE3G fragment of pLenti-CaMKII-ArchT3.0-EYFP was prepared with the XhoI-BamHI restriction sites. The XhoI sites of the TRE3G fragment were blunt-ended with the Klenow fragment of *Escherichia coli* DNA polymerase I. The resulting TRE3G fragment was subcloned into the pAAV-CaMKII-ArchT3.0-EYFP plasmid at the MluI-BamHI restriction sites, of which the MluI site was blunt-ended, generating a pAAV-TRE3G-ArchT3.0-EYFP plasmid. The recombinant AAV vector was produced using the minimal purification method[47]. Six days after the plasmids had been transfected to AAV293 cells, the culture supernatant was collected into a 50 ml tube and then centrifuged at 700 × g for 10 min by centrifuge with swing-rotor (KN-70, level = 7, KUBOTA, Tokyo, Japan). The supernatant was filtered through a Millex-HV 0.45 µm syringe-filter (SLHV033R, Merck Millipore, Darmstadt, Germany) into a new 50 ml tube. The filtered supernatant was poured into the optimized VIVASPIN filter unit (No. VS2041, Sartorius Stedim Lab Ltd., Stonehouse, UK) and was centrifuged at 1600 × g until reaching less than 1 ml repeatedly for 20–30 min at room temperature (KN-70, level = 9, KUBOTA, Tokyo, Japan). On reaching 1 ml, 15 ml of phosphate-buffered saline (PBS) was added to the filter unit with the remaining supernatant and was centrifuged at 1600 × g until reaching less than 1 ml. This step was repeated twice using fresh PBS. Finally, the recombinant AAV vector was dispensed into each tube at an optimum volume and stored at −80 °C until before starting the experiment. The recombinant AAV vector was injected with a viral titer of $4.4 \times 10^{13}$ vg/mL for AAV9-TRE-ArchT3.0-EYFP.

**Behavioral procedures**. All behavioral experiments were performed during the light cycle. Male mice were numbered and randomly assigned to an experimental group before the experiment. All behavioral experiments were performed by a researcher who was blinded to the experimental conditions. For all behavioral procedures, animals in their home cages were moved to a rack in a resting room next to the behavioral testing room and left for at least 30 min before each behavioral experiment. Different carriers were used to transfer animals from the resting

room to the behavioral testing room for every behavioral test session. After the completion of all pharmacological and optogenetic experiments, the injection sites were histologically verified. Data from animals were excluded if the animals showed abnormal behavior after surgery, such as torticollis or hair pulling, or if remarkable weight loss was observed, the target area was missed, or the bilateral expression of the virus was inadequate (Supplementary Tables 1 and 2, excluded).

*The context pre-exposure and IS task*. A previous procedure[23] with minor modifications was employed. Two contexts (context A and context B) were used in this study. Context A was a cylindrical environment (diameter × height: 180 × 230 mm) with a beige floor and a wall covered with black tape. Context B was a plexiglass front and gray side- and back-walls (width × depth × height: 175 × 165 × 300 mm), and the floor consisted of stainless steel rods that were connected to a shock generator (Muromachi Kikai, Tokyo, Japan).

Mice were placed in context A (unpaired) or context B (paired) for 6 min (context pre-exposure), and then returned to their home cage. Thirty min or 1 day after context pre-exposure, the mice were given a 0.8 mA foot shock for 2 s (IS) in context B, 5 s after the acclimation, and returned to their home cage 1 min after the IS. After 1 day, mice were placed back into context B to test their fear memory for 3 min. Context re-exposure for 3 min was repeated as reactivation, test 1, test 2, and test 3, as shown in Figs. 4 and 5. At the end of each session, mice were returned to their home cages and the chambers were cleaned with 70% ethanol. All experiments were conducted using a video tracking system (Muromachi Kikai, Tokyo, Japan) to automatically measure freezing behavior. Freezing was defined as a complete absence of movement, except for respiration. We started scoring the duration of the freezing response after 1 s of sustained freezing behavior. For each test, the freezing percentage was then averaged across mice within the same group.

*The context memory test*. The context memory test was carried out as described previously[26]. Mice were placed in context B for 6 min and were then returned to their home cages. After 1 day, mice were placed back into context B for 3 min to test their context memory, which was measured using motility. Motility was measured using a video tracking system (Muromachi Kikai, Tokyo, Japan), and was calculated as the cumulative area of movement (pixel size) per 0.1 s in the conditioning and test sessions.

*The social interaction test*. Test mice were placed into new cages prior to the experimental session and allowed to habituate to the new environment for 60 min. A 6- to 8-week-old male C57BL/6J mouse that had been anesthetized with an intraperitoneal injection of pentobarbital (80 mg/kg of body weight) was placed

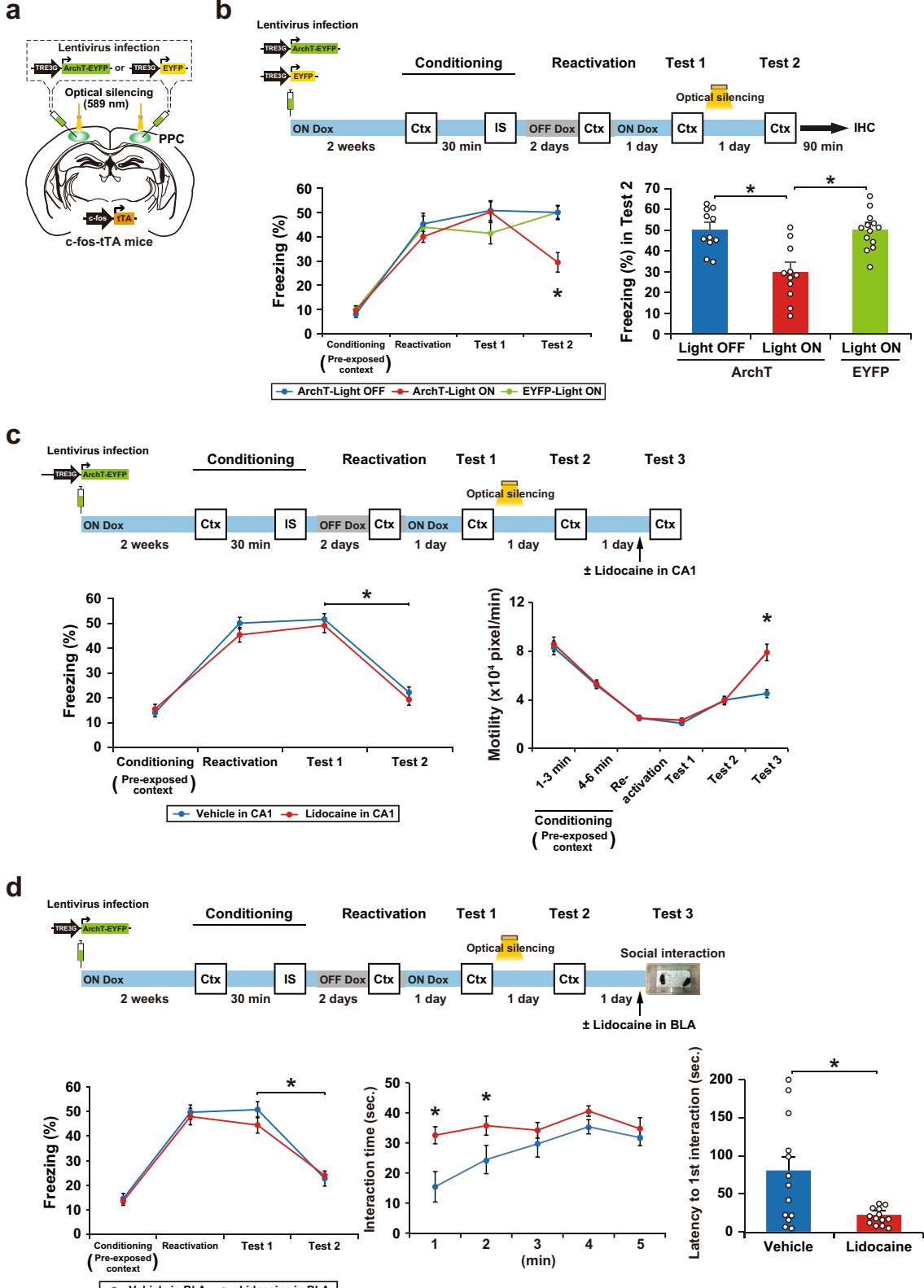

into the cage with the subject for 5 min. Social behavior was assessed by measuring the interaction time and the latency to the first interaction of test mice to anesthetized mice using a hand-held stopwatch. The interaction was defined as direct contact between the test mouse's nose and the body of the anesthetized mouse.

**Stereotaxic virus injection.** The c-fos-tTA transgenic mice were maintained with food containing 40 mg/kg Dox from weaning until the age of 12–18 weeks, at the time of surgery. The male C57BL/6J and c-fos-tTA transgenic mice were

anesthetized with a pentobarbital solution (80 mg/kg of body weight; intraperitoneal injection), and the fully anesthetized mice were placed in a stereotactic apparatus (David Kopf Instruments, Tujunga, CA, USA). The male C57BL/6J and c-fos-tTA transgenic mice were injected with LV or AAV to the PPC, CA1, or BLA. The positions (in mm) were as follows: (i) −2.0 AP, ±1.5 ML from bregma, −0.5 DV from the brain surface for the PPC; (ii) −2.0 AP, ±1.5 ML, −1.5 DV from bregma for the CA1; (iii) −1.5 AP, ±3.3 ML, −4.9 DV from bregma for the BLA. Borosilicate glass capillaries (G100T-4, outer diameter: 1.0 mm; inner diameter:

**Fig. 5 Optical silencing of PPC ensemble activity dissociates the pre-exposed context-IS association without affecting original memories. a** Labeling of PPC neurons in c-fos-tTA transgenic mice with the LVs TRE3G-ArchT-EYFP or TRE3G-EYFP. **b** The behavioral experiment with optical silencing (top). Blue and gray bars indicate the presence or absence of Dox, respectively. Optical silencing (15 min) to the PPC was delivered in the home cage immediately after test 1. The left graph shows the freezing level at conditioning, reactivation, test 1, and test 2 (bottom) (ArchT-Light OFF, $n = 11$ mice, ArchT-Light ON, $n = 11$ mice, EYFP-Light ON, $n = 13$ mice) (two-way ANOVA with RM followed by Sidak's post hoc test, session: $P < 0.0001$, group: $P = 0.0939$, session × group: $P = 0.0008$). The right graph shows the freezing level of each group at test 2 (one-way ANOVA followed by Tukey's post hoc tests, $P < 0.0001$). Mice were sacrificed 90 min after test 2 for immunohistochemistry (Fig. 6e, f). **c** The behavioral experiment with 15 min optical silencing (top). Optical silencing to the PPC was delivered in the home cage immediately after test 1. Lidocaine was injected in the CA1 region 15 min before test 3. The graph shows the freezing level at conditioning, reactivation, test 1, and test 2 (left), and motility during the conditioning, reactivation, test 1, test 2 and test 3 (right) (Vehicle in CA1, $n = 16$ mice, Lidocaine in CA1, $n = 17$ mice) (two-way ANOVA with RM followed by Sidak's post hoc test, freezing; session: $P < 0.0001$, group: $P = 0.3449$, session × group: $P = 0.4925$, motility; session: $P < 0.0001$, group: $P = 0.0322$, session × group: $P < 0.0001$). **d** The behavioral experiment with 15 min optical silencing (top). Optical silencing to the PPC was delivered in the home cage immediately after test 1. Lidocaine was injected in the BLA 15 min before test 3. The graph shows the freezing level at conditioning, reactivation, test 1, and test 2 (left), the interaction time (center) and the latency to the first interaction (right) in test 3 (bottom) (Vehicle in BLA, $n = 13$ mice, Lidocaine in BLA, $n = 13$ mice) (two-way ANOVA with RM followed by Sidak's post hoc test, Freezing; session: $P < 0.0001$, group: $P = 0.4608$, session × group: $P = 0.4670$, interaction time; time: $P = 0.0028$, group: $P = 0.0003$, time × group: $P = 0.2251$, two-tailed unpaired $t$ test, latency to 1st interaction; $P = 0.0060$). Error bars represent the mean ± s.e.m. *$P < 0.05$. Ctx context, IS immediate shock, IHC immunohistochemistry. For details of statistical data, see Supplementary Table 1. Source data are provided as a Source Data file.

0.78 mm; Warner Instruments, Hamden, CT) were pulled using a Micropipette Puller (P-1000, Sutter Instrument, Novato, CA, USA; heat: 778, pull: 25, vel: 150, time: 250, and pressure: 500). The tip of the glass capillary was trimmed with scissors under an operating microscope to have an outer diameter of about 50 µm. For the virus delivery, 1 µl LV (delivery rate, 0.1 µl/min) or 0.3 µl AAV (delivery rate, 0.06 µl/min) was injected using a motorized stereotaxic injector (Narishige, Tokyo, Japan) and a Hamilton microsyringe coupled with a glass capillary. After the injection, the capillary was left in place for 10 min and then slowly withdrawn. To insert an optical fiber into the PPC, CA1, BLA, or ACC, guide cannulas (C316GS-4/SPC, Plastics One, Roanoke, VA, USA) were implanted slightly above the target coordinates of each region. The coordinate for the ACC (in mm) was +1.0 AP, +0.5 ML from bregma, −0.5 DV from the brain surface. The guide cannulas were fixed on the skull using dental cement. A dummy cannula with a cap was used to protect the guide cannula from clogging up. After recovery from anesthesia in a stable-temperature incubator, mice were returned to their home cages.

**Optic fiber placement and optical stimulation.** Optical fiber unit placement was carried out as described previously[8,10]. For the placement of the optical fiber units, mice were anesthetized with 2.0% isoflurane and the dummy cannulas were removed from the guide cannulas. The optical fiber unit, composed of a plastic cannula body, was a two-branch-type unit with an optic fiber diameter of 0.250 mm (COME2-DF2-250, Lucir, Tsukuba, Japan). The optical fiber unit was inserted into the guide cannula, and the bodies of the guide cannula and the optical fiber unit were tightly connected using an adhesive agent (Kokuyo, Osaka, Japan). The tips of the optical fibers were targeted slightly above the target coordinates of each region. The fiber unit connected with the mouse was attached to an optical swivel (COME2-UFC, Lucir, Tsukuba, Japan), which was connected to a laser (200 mW, 473 nm and 589 nm, COME-LB473/589/200, Lucir, Tsukuba, Japan) via a main optical fiber. The delivery of light pulses was controlled by a schedule stimulator (COME2-SPG-2, Lucir, Tsukuba, Japan) in time-lapse mode. For the silencing of cell assemblies by ArchT or NpHR, optical illumination (continuous 589 nm light, 12–14 mW at the fiber tip) was delivered, as shown in each figure, except that optical illumination in the IS session was delivered 1 min before starting. For the activation of cell assemblies by ChR2, optical illumination (473 nm light, 20 Hz, 10 ms, 6–7 mW at the fiber tip) was delivered. Two hours after the optical illumination had ended, mice were anesthetized with 2.0% isoflurane, the optic fiber unit was detached, and the mice were returned to their home cage. All mice were sacrificed and immunostained to verify the coordinates after behavioral experiments. Mice in which no opsin expression was confirmed were excluded from each experiment (Supplementary Tables 1 and 2, excluded).

**Drug infusion.** Mice anesthetized with pentobarbital (80 mg/kg of body weight) were implanted bilaterally with stainless steel guide cannulas (C316GS-4/SPC, Plastics One, Roanoke, VA, USA) using the following stereotactic coordinates (in mm): (i) −2.0 AP, ± 1.5 ML from bregma, −0.3 DV from the brain surface for injection into the PPC; (ii) −2.0 AP, ±1.5 ML, −0.5 DV from bregma for the CA1 region; (iii) −1.5 AP, ±3.3 ML, −3.4 DV from bregma for the BLA. After surgery, a dummy cannula was inserted into the guide cannula, and mice were allowed to recover for at least 14 days in individual home cages. To prevent disruption of the target region, the injection cannula (C316IS-4/SPC, Plastics One, Roanoke, VA, USA) extended beyond the tip end of the guide cannula by 0.2 mm for the PPC, and 1.0 mm for the CA1 and the BLA. A sodium channel blocker (lidocaine-hydrochloride; Sigma, St. Louis, MO, USA, L5647) was used to inactivate the PPC, CA1 region, or BLA. Mice were briefly anesthetized with isoflurane to facilitate the

insertion of the injection cannula. Lidocaine-hydrochloride (4%, 1 µl) or PBS was infused into each brain region at a rate of 0.333 µl/min, which was controlled by a microsyringe pump (Legato111, KD Scientific Inc., Holliston, MA, USA). After infusion, the injection cannula was left in place for 3 min to allow for drug diffusion. After behavioral experiments, all mice were injected 0.5 µl of 0.5 µM Rhodamine B (Sigma, St. Louis, MO, USA) through the guide cannula to verify the coordinates. Mice in which no Rhodamine B expression was confirmed were excluded from each experiment (Table S2, excluded).

**Tracer injection.** BDA (Thermo Fisher Scientific, Waltham, MA, USA, D1817) and fluorescent retrograde tracer cholera toxin B subunit conjugated to fluorophore Alexa 488 (CTB488, Thermo Fisher Scientific, Waltham, MA, USA, C34775) were used to label neurons that projected from the target regions. BDA and CTB488 were unilaterally delivered into the PPC and ACC of male C57BL6J mice using a glass micropipette at the following coordinates (in mm): (i) −2.0 AP, +1.5 ML from bregma, −0.5 DV from the brain surface for the PPC; (ii) +1.0 AP, +0.5 ML from bregma, −0.5 DV from the brain surface for the ACC. Mice were injected with 200 nl of 10% BDA in the PPC or 200 nl of 1% CTB488 in the ACC, and mice were perfused 7 days or 5 days after injection. BDA was visualized by staining with streptavidin-Alexa Fluor 555 (1:1000, Life Technologies, Carlsbad, CA, USA).

**Fluorescent in situ hybridization.** Five minutes after the behavioral experiments, brains were rapidly extracted, frozen, and stored at −80 °C before sectioning. Twenty micrometer-thick coronal sections of mouse brain were cut using a cryostat, air-dried, and mounted onto slides.

Fluorescein-labeled arc riboprobes were synthesized using a transcription kit (MaxiScript, Ambion Inc., Austin, TX, USA) and premixed RNA labeling nucleotide mixes containing fluorescein-labeled UTP (Roche Diagnostics, Indianapolis, IN, USA). Riboprobes were purified using G-50 spin columns (GE Healthcare, Waukesha, WI, USA). The plasmid containing nearly full-length cDNA of the arc transcript was used to generate probes.

Fluorescent in situ hybridization (FISH) was carried out as described previously[8,10,23]. For Arc CatFISH, brain sections were fixed in fresh 4% paraformaldehyde/0.1 M phosphate buffer at room temperature. All steps were performed at room temperature unless otherwise indicated. Sections were acetylated with 0.5% acetic anhydride/1.5% triethanolamine, dehydrated using 1:1 methanol/acetone solutions, and equilibrated in 2× saline sodium citrate buffer (SSC). Sections were then prehybridized for 30 min in prehybridization buffer [50% formamide, 5% sheared salmon sperm DNA (Sigma, St. Louis, MO, USA), 0.3% yeast tRNA (Sigma, St. Louis, MO, USA), 5× SSC, 5× Denhart's solution (Sigma, St. Louis, MO, USA)]. The antisense riboprobe was diluted in hybridization buffer at a final concentration of 1.5 µg/ml and applied to each slide. Coverslips were added to the slides, and hybridization was carried out at 56 °C for 16 h. Sections were dipped in 2× SSC and rinsed in 0.5× SSC followed by 0.5× SSC at 56 °C, which included an earlier wash step in 2× SSC with RNase A (10 µg/ml) at 37 °C. Sections were again washed in 0.5× SSC and endogenous peroxidase activity was inhibited by incubation in 2% $H_2O_2$ in 1× SSC for 15 min. Sections were blocked with TSA blocking reagent (PerkinElmer, Waltham, MA, USA) and incubated with an anti-fluorescein horseradish peroxidase conjugate (1:500, PerkinElmer, Waltham, MA, USA) overnight at 4 °C. Sections were washed three times in Tris-buffered saline (with 0.05% Tween-20), and the conjugate was detected using the Fluorescein TSA Plus System (1:10, PerkinElmer, Waltham, MA, USA) for 2 h. Sections were washed in PBS and the nuclei were counterstained with DAPI (1 µg/ml, Roche

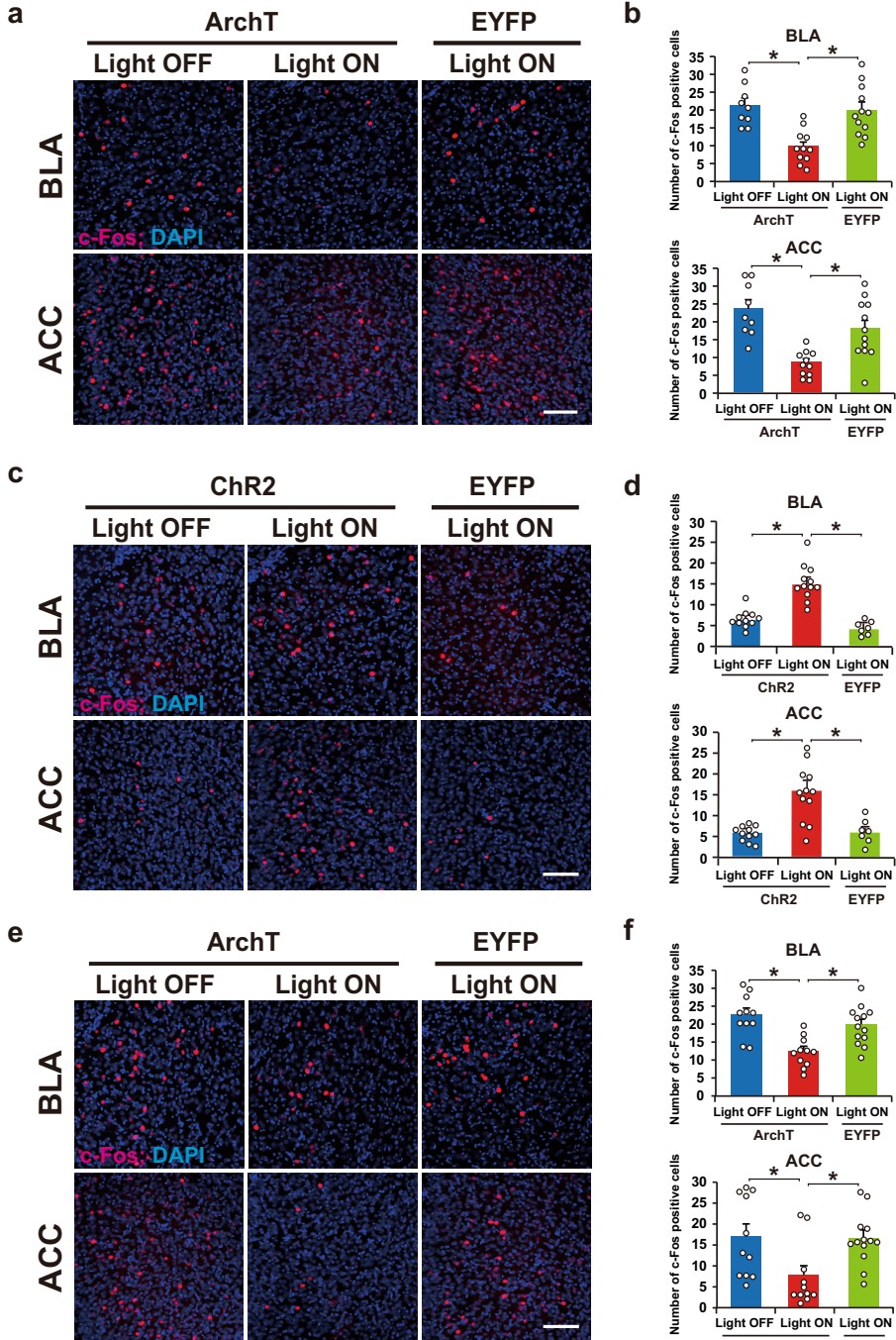

**Fig. 6 c-Fos-positive cells in the BLA and the ACC after the test session. a, c, e** Representative images of c-Fos expression 90 min after the test shown in Fig. 2b (panel **a**), Fig. 3b (panel **c**), and Fig. 5b (panel **e**). Scale bar, 100 μm. **b, d, f** Quantitative proportion of c-Fos-positive cells in the BLA (top) and ACC (bottom) (panel **b**: ArchT-Light OFF, $n = 9$ mice, ArchT-Light ON, $n = 11$ mice, EYFP-Light ON, $n = 12$ mice; panel **d**: ChR2-Light OFF, $n = 11$ mice, ChR2-Light ON, $n = 12$ mice, EYFP-Light ON, $n = 7$ mice; panel **f**: ArchT-Light OFF, $n = 11$ mice, ArchT-Light ON, $n = 11$ mice, EYFP-Light ON, $n = 13$ mice.) (one-way ANOVA followed by Tukey's post hoc tests, panel **b**: BLA; $P = 0.0002$, ACC; $P < 0.0001$, panel **d**: BLA; $P < 0.0001$, ACC; $P = 0.0002$, panel **f**: BLA; $P = 0.0001$, ACC; $P = 0.0120$). Error bars indicate the mean ± s.e.m. *$P < 0.05$. For details of statistical data, see Supplementary Table 1. Source data are provided as a Source Data file.

Diagnostics, Indianapolis, IN, USA, 10236276001) for 20 min. Finally, the sections were washed with PBS and mounted with Permafluor mounting medium (Lab Vision Corporation, Fremont, CA, USA).

Images were acquired using a Zeiss LSM 780 confocal microscope with a 40×/1.2 NA objective lens. PMT assignments, pinhole sizes, and contrast values were kept constant. Images of the CA1 region and BLA were acquired by collecting z-stacks (1 μm-thick optical sections). Using the Zen software, each cell was characterized through several serial sections, and only cells containing whole nuclei were included in the analysis. The details of the classification analysis have been

described before[24]. Small, bright, uniformly DAPI-stained nuclei (from putative glial cells) were not analyzed. All other whole nuclei were analyzed from top to bottom. The designation "cytoplasm-positive (Cyto) neurons" was given to cells containing perinuclear/cytoplasmic labeling in multiple optical sections. The designation "nuclear-positive (Nuc) neurons" was given to cells containing two small, intense intranuclear fluorescent foci, and the designation "Nuc and Cyto double-positive (Cyto/Nuc)" was given to cells containing both intranuclear and cytoplasmic Arc-positive signals. Three sections corresponding to each region of interest (ROI) were chosen from each mouse.

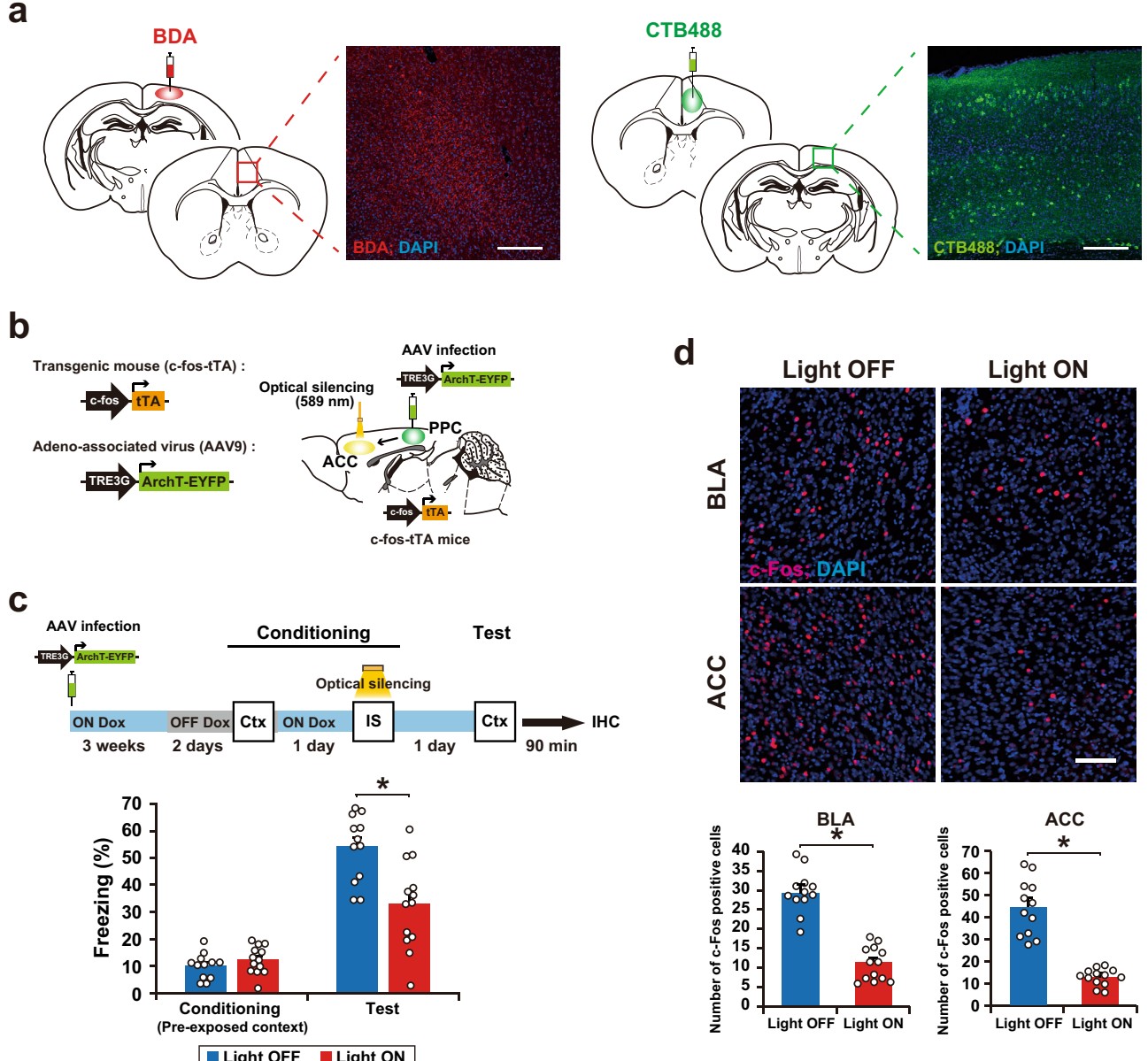

**Fig. 7 A PPC–ACC circuit regulates memory association. a** Horizontal brain sections of the ACC from a mouse injected with BDA in the PPC, in which neurons were immunostained with streptavidin-Alexa Fluor 555 (red; left). PPC cell axons were detected in the ACC. Horizontal brain sections of the PPC from a mouse injected with CTB488 in the ACC (green; right). CTB488-labeled cells were detected in the PPC. Scale bar, 200 μm. These immunohistochemical analyses were repeated at least three times independently with similar results. **b** Schematics showing labeling and manipulation of the PPC–ACC circuit in c-fos-tTA transgenic mice with the AAV9 TRE3G-ArchT-EYFP. **c** Silencing of cell ensembles during an IS session of conditioning (top). Blue and gray bars indicate the presence or absence of Dox, respectively. Mice were sacrificed 90 min after the test for immunohistochemistry (panel **d**). The graph shows the freezing level at the conditioning and test session (bottom) (Light OFF, $n = 12$ mice, Light ON, $n = 13$ mice) (two-way ANOVA with RM followed by Sidak's post hoc test, session: $P < 0.0001$, group: $P = 0.0100$, session × group: $P = 0.0003$). **d** Representative image of c-Fos expression 90 min after the test (top). Scale bar, 100 μm. The proportion of c-Fos-positive cells in the BLA (left) and ACC (right) (Light OFF, $n = 12$ mice, Light ON, $n = 13$ mice) (two-tailed unpaired $t$-test, BLA; $P < 0.0001$, ACC; $P < 0.0001$). Error bars represent the mean ± s.e.m. *$P < 0.05$. Ctx context, IS immediate shock, IHC immunohistochemistry. For details of statistical data, see Supplementary Table 1. Source data are provided as a Source Data file.

**Immunohistochemistry**. Mice were deeply anesthetized with an overdose of pentobarbital solution and perfused transcardially with PBS, pH 7.4, followed by 4% paraformaldehyde in PBS. The brains were removed and further post-fixed by immersion in 4% paraformaldehyde in PBS for 24 h at 4 °C. Each brain was equilibrated in 30% sucrose in PBS and then frozen in dry ice powder. For EYFP and c-Fos staining, coronal sections were cut on a cryostat at a thickness of 50 μm, 1.2 mm to 0.8 mm, and −1.4 to −1.8 mm from bregma for the ACC and BLA, respectively, and transferred to 12-well cell culture plates (Corning, Corning, NY, USA) containing PBS. After washing with PBS, the floating sections were treated with blocking buffer, 5% normal donkey serum (S30, Chemicon by EMD Millipore, Billerica, MA, USA) in PBS with 0.2% Triton X-100 (PBST), at room temperature

for 1 h. Reactions with primary antibodies were performed in blocking buffer containing rabbit anti-GFP (1:1000, Invitrogen, Carlsbad, CA, USA, A11122) and/ or goat anti-c-Fos (1:1000, SantaCruz, Santa Cruz, CA, USA, SC-52G) antibodies for double staining of EYFP and c-Fos, chicken anti-GFP (1:1000, Abcam, Cambridge, UK, ab13970), and/or rabbit anti-Egr-1 (1:1000, SantaCruz, Santa Cruz, CA, USA, SC-189) antibodies for double staining of EYFP and Zif268 at 4 °C for 1 or 2 days. After three 10 min washes with PBS, the sections were incubated with donkey anti-rabbit IgG-Alexa Fluor 488 (1:300, Life Technologies, Carlsbad, CA, USA, A21206) and/or donkey anti-goat IgG-Alexa Fluor 546 (1:300, Life Technologies, Carlsbad, CA, USA, A11056) secondary antibodies for double staining of EYFP and c-Fos, and donkey anti-chicken IgG-Alexa Fluor 488 (1:300, Jackson

ImmunoResearch Laboratories, Inc, West Grove, PA, USA, 703-545-155) and donkey anti-rabbit IgG-Alexa Fluor 546 (1:300, Life Technologies, Carlsbad, CA, USA, A10040) secondary antibodies for double staining of EYFP and Zif268 at room temperature for 3 h. Sections were treated with DAPI (1 μg/ml, Roche Diagnostics, Indianapolis, IN, USA, 10236276001) and then washed with PBS three times, 10 min/wash. Mounting of sections on glass slides was performed with ProLong Gold antifade reagents (Invitrogen, Carlsbad, CA, USA). Images were acquired using a Zeiss LSM 780 confocal microscope with a 20×/1.2 NA objective lens. PMT assignments, pinhole sizes, and contrast values were kept constant. For quantification of the number of positive cells, images of the ROI were acquired by collecting z-stacks (1 μm-thick optical sections). Quantification of the number of c-Fos-positive or Zif268-positive cells in the ROI was performed by an experimenter who was blinded to the experimental conditions.

**Statistics and reproducibility**. Statistical analysis was performed using GraphPad Prism version 6 (GraphPad Software, San Diego, CA, USA). All data are presented as the mean ± SEM. The number of animals used is indicated by "$n$". Comparisons between two groups were made using two-tailed unpaired Student's $t$-tests. Multiple group comparisons were assessed using a one-way, two-way, or repeated-measures analysis of variance (ANOVA), followed by the Tukey–Kramer or Sidak's post hoc test when significant main effects or interactions were detected. The null hypothesis was rejected at the $P < 0.05$ level. All statistical data are described in figure legends and Supplementary Tables 1 and 2. All experiments were repeated at least two times independently with similar results.

**Reporting summary**. Further information on research design is available in the Nature Research Reporting Summary linked to this article.

## Data availability
The data generated in this study are provided in the Supplementary Information/Source Data files. Source data are provided with this paper. Further information will be available from the corresponding author on reasonable request. Source data are provided with this paper.

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

## Acknowledgements
We thank K. Deisseroth (Stanford University) for the ArchT 3.0-EYFP cDNA and ChR2(T159C)-EYFP cDNA; H. Hioki and T. Kaneko (Kyoto University) for the TGB vector; H. Nomura (Hokkaido University) for guidance with CatFISH method; H. Nishizono, M. Matsuo, and Y. Arasaki-Yanagihashi for generating the transgenic mice; and S. Tsujimura and S. Okami for maintaining the transgenic mice. All members of the Inokuchi laboratory supported and discussed this study. For histology and targeting figures we used The Mouse Brain atlas as a reference for the generation of brain slice images (Paxinos G., Franklin K.B.J. 2001. The mouse brain in stereotaxic coordinates. 2nd Ed. Elsevier

Academic Press). This work was supported by the Core Research for Evolutional Science and Technology (CREST) program (JPMJCR13W1) of the Japan Science and Technology Agency (JST), JSPS KAKENHI grant numbers JP18H05213 and JP23220009, a Grant-in-Aid for Scientific Research on Innovative Areas "Memory dynamism" (JP25115002) from MEXT, and the Takeda Science Foundation to K.I.; JSPS KAKENHI JP 24680034, the Hokuriku Bank Grant for Young Scientists, The Uehara Memorial Foundation, The Takeda Science Foundation, The Kanae Foundation for the promotion of medical science, and the Tamura Science and Technology Foundation to A.S.; and Brain Mapping by Integrated Neurotechnologies for Disease Studies (Brain/MINDS) from Japan Agency for Medical Research and development, AMED (JP20dm0207057) to H.H.

## Author contributions

A.S. and K.I. designed the study. A.S., S.K., and E.M. cloned all vector constructs. S.K., E.M., A.K., and H.H. prepared the viruses. N.O. introduced optogenetics techniques. A.S., S.K., E.M., and E.S performed all animal surgery, behavioral experiments, histology, and data analyses. A.S. and K.I. wrote the paper and contributed to the interpretation. K.I. supervised the entire project. All authors discussed and commented on the paper.

## Competing interests

The authors declare no competing interests.
