## [Peer Review File · Nature Communications]

A cortical cell ensemble in the posterior parietal cortex controls past experience-dependent memory updatingEditorial Note: This manuscript has been previously reviewed at another journal that is not operating a transparent peer review scheme. This document only contains reviewer comments and rebuttal letters for versions considered at *Nature Communications*.

REVIEWERS' COMMENTS

Reviewer #1 (Remarks to the Author):

The authors have addressed my concerns and I believe this paper is suitable for publication in nature communications. One minor issue though is that I do think the authors should provide one line in the results or the discussion indicating that in the IS-induced social interaction test its not clear if you labeled a random % of neurons in BLA outside of the IS, if inhibition of these cells would also rescue IS-induced anxiety. If that was the case then either IS memory is stored in BLA, or any inhibition of BLA can produce reduction the same change in behavior that is interpreted as "IS memory is stored in the BLA"

Responses to Reviewers' comments

Reviewer #1:

The authors have addressed my concerns and I believe this paper is suitable for publication in nature communications. One minor issue though is that I do think the authors should provide one line in the results or the discussion indicating that in the IS-induced social interaction test its not clear if you labeled a random % of neurons in BLA outside of the IS, if inhibition of these cells would also rescue IS-induced anxiety. If that was the case then either IS memory is stored in BLA, or any inhibition of BLA can produce reduction the same change in behavior that is interpreted as "IS memory is stored in the BLA"

Authors' response:

Thank you for your valuable comment. As reviewer suggested, we cannot exclude the possibility that inhibition of randomly labeled BLA neurons would also rescue the anxiety induced by IS. We revised as follows.

“This result strongly suggests that the inhibition of IS memory recall caused the suppression of anxiety, although it is unclear whether random inhibition of BLA neurons would rescue IS-induced anxiety.”

We have described this in the Result on page 7, line 23 to page 8, line 1.